# Molecular Identification and Mycotoxin Production by *Alternaria* Species Occurring on Durum Wheat, Showing Black Point Symptoms

**DOI:** 10.3390/toxins12040275

**Published:** 2020-04-23

**Authors:** Mario Masiello, Stefania Somma, Antonia Susca, Veronica Ghionna, Antonio Francesco Logrieco, Matteo Franzoni, Stefano Ravaglia, Giuseppe Meca, Antonio Moretti

**Affiliations:** 1Institute of Sciences of Food Production, National Research Council (CNR-ISPA), Via Amendola 122/O, 70126 Bari, Italy; stefania.somma@ispa.cnr.it (S.S.); antonella.susca@ispa.cnr.it (A.S.); veronica.ghionna@ispa.cnr.it (V.G.); antonio.logrieco@ispa.cnr.it (A.F.L.); 2S.I.S. Società Italiana Sementi S.p.A, Via Mirandola 1, 40068 San Lazzaro di Savena (BO), Italy; teofranzo@gmail.com (M.F.); s.ravaglia@sisonweb.com (S.R.); 3Department of Preventive Medicine, Nutrition and Food Science Area, University of Valencia (Spain), Avenida Vicent Andres Estelles s/n, Burjassot, 46100 Valencia, Spain; giuseppe.meca@uv.es

**Keywords:** alternariol, alternariol-monomethyl ether, tenuazonic acid, altenuene, section *Alternaria*, section *Infectoriae*, species specific-primers

## Abstract

Black point is a fungal disease of wheat, mainly associated with mycotoxigenic *Alternaria* species. Affected wheat kernels are characterized by dark brown discolouration of the embryo region and reduction of grain quality. Potential risk is the possible accumulation of *Alternaria* mycotoxins, alternariol (AOH), alternariol-monomethyl ether (AME), tenuazonic acid (TA), and altenuene (ALT), provided by haemato-toxic, genotoxic, and mutagenic activities. One hundred and twenty durum wheat samples belonging to 30 different genotypes grown in Bologna and Modena areas, in Italy, showing black point symptoms, were analyzed for *Alternaria* species and their mycotoxin contamination. Alternariol was selected as an indicator of the capability of the *Alternaria* species to produce mycotoxin in vivo in field conditions. The data showed that *Alternaria* species occurred in 118 out of 120 wheat kernels samples, with the incidence of infected kernels ranging between 1% and 26%. Moreover, AOH was detected by using a HPLC with a diode array detector (LC-DAD) in 98 out of 120 samples with values ranging between 24 and 262 µg Kg^−1^. Ninety-two *Alternaria* representative strains, previously identified morphologically, were identified at species/section level using gene sequencing, and therefore were analyzed for their mycotoxin profiles. Eighty-four strains, phylogenetically grouped in the *Alternaria* section, produced AOH, AME, and TA with values up to 8064, 14,341, and 3683 µg g^−1^, respectively, analyzed by using a LC-DAD. On the other hand, eight *Alternaria* strains, included in *Infectoriae* Section, showed a very low or no capability to produce mycotoxins.

## 1. Introduction

Wheat is a largely cultivated crop worldwide in temperate areas, and is a stable food for humans, and also a basic component of the diet of livestock. In Italy, wheat has a production of about 8 million tons of durum (*Triticum durum*) and soft (*Triticum aestivum*) wheat, on a surface of about 1.8 million hectares [1].

Fungi are responsible for major wheat diseases such as black point, a worldwide disease of kernels, caused by a complex of species including mainly *Alternaria* species and *Cochliobolus sativus* [2,3,4]. Moreover, other fungi belonging to *Aspergillus*, *Cladosporium*, *Curvulavia*, *Fusarium*, *Penicillium* and *Stemphylium* genera can participate in the disease complex and all together these species can induce the expression of the symptoms of black point [5,6,7].

Among these genera, *Fusarium* and *Alternaria* associated with wheat kernels, are especially worrisome since they are able to produce mycotoxins [7,8], toxic secondary metabolites that can be accumulated in colonized tissues [9]. Indeed, the mycotoxin contamination of wheat is one of the most important issue for human and animal health at the worldwide level, since, at harvest, a wide range of toxigenic *Alternaria* and *Fusarium* species can accumulate several mycotoxins in the kernels. Such infections, beside causing productive and economic losses with serious adverse impacts on the farmer’s incomes, could lead to a serious risk for human and animal consumption, due to their toxic effects on several biological activities [10,11].

*Alternaria* is a ubiquitous and abundant fungal genus widespread in the atmosphere as well as in soil, seeds, and agricultural commodities. It includes plant pathogenic and saprophytic species that may affect crops in the field or can cause harvest and postharvest decay of plant products [9].

The main *Alternaria* species reported as associated with black point disease are *A. alternata*, *A. tenuissima*, *A. arborescens*, *A. infectoria* [4,7,12,13,14]. Moreover, at a lower frequency, minor *Alternaria* species belonging to *Infectoriae* [15,16] and *Pseudoalternaria* sections [17] were also detected on wheat [7,18,19].

Overall, the identification at species level is very difficult in this genus since most *Alternaria* species exhibit a considerable morphological plasticity that is dependent upon cultural conditions, substrate, temperature, light, and humidity [20]. In addition, several isolates have intermediate species traits that do not clearly segregate into recognized species [21,22,23].

Therefore, for a correct identification at the species level, a poly-phasic approach, based not only on morphology, but also on mycotoxin profile, and molecular characterization is worthwhile [24]. Among the most common *Alternaria* mycotoxins, there are the dibenzopyrone derivatives alternariol (AOH), alternariol mono-methyl ether (AME) and altenuene (ALT) from one side; the tetramic acid derivative Tenuazonic Acid (TA), from the other.

The toxicological risk of AOH and AME has been elucidated by Da Cruz Cabrala et al. [25] that demonstrated their genotoxic activity on human cell lines of colon cancer. In addition, TA induces the human hematologic disorder called “onalay” and precancerous changes in the esophageal mucosa of mice [26]. Finally, for altenuene, acute toxicity has been reported [27].

Several studies reported the occurrence of *Alternaria* species on cereals worldwide, and the capability of *Alternaria* strains to synthetize wide contents of mycotoxins in in vitro media, but few studies reported the occurrence of mycotoxins on wheat and cereal foodstuffs. Recently, Romero Bernal et al. [28] reported high AOH, AME and TA contaminations of wheat and wheat derivatives, in Argentina, among which TA was the main mycotoxin detected. In a previous study, Azcarate et al. [29] reported that about 20% of Argentinian wheat samples were highly contaminated with *Alternaria* toxins, with a significant AOH content. On the other hand, a natural occurrence of ALT, AME and TA in wheat samples collected in different areas of China, Sweden and Albania was reported [30,31,32]. In addition, recently, AOH has been described as a virulence and colonization factor during apple, citrus, and tomato infection by *Alternaria alternata* [33]. Therefore, to detect the presence of *Alternaria* toxins in wheat and accurately identify the species that produce them is an important step for better evaluating not only the toxigenic risk associated with infected kernels, but also to understand if such metabolites are produced in the field during the infection process.

Although the genus *Alternaria* is considered an important wheat colonizer able to produce the abovementioned mycotoxins in vivo, the occurrence of *Alternaria* mycotoxins in cereals has been largely neglected for long time. This is probably due also to the lack of legislation on *Alternaria* mycotoxin occurrence in commodities. On the other hand, very poor studies have been devoted to evaluate the susceptibility of wheat to black point disease, have for many years considered only a problem for the quality and nutritive decline of wheat products, and not possibly related to safety. 

Therefore, the present work mainly aimed to: (a)Identify the main fungal species infecting kernel wheat samples showing black point symptoms, collected in Italy;(b)Detect the presence of AOH in wheat kernels in the field;(c)Study the genetic variability of the *Alternaria* species isolated, by sequencing the genes coding for allergen alt a1 (*alt-a1*), glyceraldeyde-3-phosphate dehydrogenase (*gpd*) and translation elongation factor 1-α (*tef*), by following a multi-locus gene sequence approach;(d)Analyze the mycotoxin profile of selected strains of the *Alternaria* species identified.

## 2. Results

### 2.1. Black Point Disease Symptoms on Wheat Samples

At harvest, black point disease symptoms were evaluated on wheat plants, in both Bologna and Modena fields, according to an empirical scale with ten classes of severity (0 = absence of infections, 9 = extensive symptoms of infection). Data on black point symptoms are reported in Table 1 and Table 2.

In the Bologna field, with the exception of the G17 genotype, exposed to fungicide treatment, all theses showed black point symptoms. Black point symptoms were more severe on plants not exposed to fungicide treatment. Indeed, comparing the same genotypes, treated or untreated with fungicide, black point symptoms were more evident on untreated plants (Table 1). On untreated plants, symptoms ranged between 0.5 (G5 genotype) and 8 (G9 genotype), with a mean value of 3.5. Treated plants showed symptoms ranging between 0 (G17) and 4 (G1 and G16 genotypes), with a mean value of 1.9. Data on commercial varieties are summarized also in Table 1 and confirmed that fungicide treatment reduced drastically black point symptoms.

Data from the Modena field are summarized in Table 2 and confirmed that the highest black point symptoms were recorded in untreated plots, although they were lower than in the Bologna field.

### 2.2. Alternariol Contamination of Wheat Samples

Data on contamination of kernels by AOH are reported in Table 1. In the Bologna field, 88% of the samples were contaminated by AOH. Three samples exposed to fungicide spraying (G5, G9, G23) and four untreated samples (G4, G8, G11, Iride) were not contaminated from AOH. In the plots exposed to fungicide treatment the contamination values ranged from 24 µg kg^−1^ (G3) to 92 µg kg^−1^ (Saragolla), with a mean value of 39 µg kg^−1^. In the untreated samples the contamination values ranged between 40 (G22) and 148 (G2) µg kg^−1^, with a mean value of 84 µg kg^−1^. With the exception of the samples G4, G8 and Iride, in which chemical treatment did not influence AOH contamination, the treated plots showed a lower contamination of AOH than untreated plots (Table 1).

In the Modena field, AOH contamination was higher than in the Bologna field. In treated plots the AOH contamination ranged between 64 (G9) and 262 (G5) µg kg^−1^, with mean value of 109 µg kg^−1^; in an untreated plot the level of AOH contamination ranged between 68 (G20) and 230 (G6) µg kg^−1^, with a mean value 66 µg kg^−1^. In the Modena trial, a strict correlation between AOH contamination and chemical treatment was not observed. In 17 out of 30 untreated samples the AOH contamination was lower than the same genotypes or commercial cultivar spayed with chemicals (Table 2).

### 2.3. Detection of Fungal Infection

Fungal infection was detected in all wheat samples, with values ranging between 21% and 83% in the Bologna field and between 71% and 98% in the Modena field (data not shown). Un-treated samples were slightly more infected than treated samples. Particular attention was focused on *Alternaria* and *Fusarium* genera. Among the fungal colonies morphologically identified as belonging to *Alternaria* and *Fusarium* genera, 382 monoconidial representative isolates of *Alternaria spp.* and 177 of *Fusarium spp.* were recovered from the infected kernels.

Infection by *Alternaria* species occurred in 118 out of 120 wheat samples, ranging from 1% to 26% in the Bologna field (Table 1), with mean value of 8% and 10% for treated and untreated theses, respectively; the incidence of infected kernels ranging between 2% and 20% were detected in Modena field (Table 2), with mean value of 8% and 12% in treated and untreated theses, respectively. The infection rate at the *Alternaria* species level was also evaluated.

In 88 out of 120 wheat samples, strains belonging to the *Fusarium* genus were also detected; however, *Fusarium* spp incidence of infected kernels was very low, up to 7%, with mean values in treated and untreated plots of 1.2% and 2.3%, respectively (data not shown).

Total fungal infection and therefore *Alternaria* infection rate was lower in treated than untreated theses, according with the black-point symptoms assessed in fields during the field inspections (Table 1; Table 2). However, between the symptoms of black point disease and average of *Alternaria* species infection, a strict correlation was not observed.

### 2.4. Morphological Characterization

All the 382 mono-conidial strains, representative of the whole population found on the 120 samples, grew very quickly on potato dextrose agar (PDA) and potato-carrot agar (PCA), usually covering the whole surface of the Petri plates, and showing abundant sporulation. Conidia and conidiophore branch morphologies were used to identify the *Alternaria* species, according to Simmons [21]. Almost all 382 isolates were related to four morphospecies: *A. alternata*, *A. tenuissima*, *A. arborescens*, *A. infectoria*. In detail, 125 isolates were identified as *A. alternata*, 152 were identified as *A. tenuissima*, 72 were identified as *A. arborescens* and 33 were identified as *A. infectoria*.

### 2.5. PCR Amplification and Phylogenetic Analysis

All the 92 selected *Alternaria* strains gave PCR products of the expected size for the *alt-a1* and *tef* genes tested. In all *Alternaria* strains, morphologically identified as *A. infectoria*, a detection of 55 nt in the first part of the *gpd* region sequence amplified, was observed.

Fragment of *tef* gene showed a lower variability with few polymorphic regions. The gene showing the higher degree of variability was *alt-a1*. In particular, the higher variability was observed in all the *alt-a1* sequences where a 6 nt deletion in *A. infectoria* group compared to *A. alternata*, *A. tenuissima* and *A. arborescens* groups was observed. Moreover, the final part of the resulting fragment was highly polymorphic.

The analysis of the combined sequences of the three genes resulting in a Maximum Parsimony tree is shown in Figure 1. The phylogenetic tree, obtained with Mega5 allowed us to define two well-separated clades, corresponding to *Alternaria* section (clade A) and *Infectoriae* section (clade B). Clade B grouped all the *Alternaria* reference strains belonging to *Infectoriae* section and eight out of 92 strains tested (Figure 1). A high variability was observed; however, seven out of eight strains of this clade and *A. ventricosa* reference strain, formed a well-supported (99 bootstrap value) group, that differed from the *A. triticina* reference strain. Among *Alternaria* strains grouped in this clade, strain ITEM17335 was highly similar to *A. rosae* E.G.S. 41–130 reference strain and was well separated by *Infectoriae* section.

A big group of 84 strains formed the Clade A, supported by a bootstrap value of 93. Within this clade, three sub-clades were defined; sub-clade A1 included 30 field strains and *A. arborescens* and *A. cerealis* reference strains. Forty-two strains were (morphologically identified as *A. alternata* and *A. tenuissima* species) grouped with *A. alternata* (E.G.S. 34.016), *A. tenuissima* (E.G.S. 34.015) reference strains, *A. alternata* BMP0270, *A. limoniasperae* BMP2335 and *A. turkisafria* BMP3436 (sub-clade A2); sub-clade A3 included 12 field strains and showed 100% homology with *A. mali* BMP3064.

In sub-clade A2, strains shared a very high level of homology and therefore it was not possible to distinguish between *A. alternata* and *A. tenuissima* species, in terms of morphological definitions. On the contrary, a great variability was observed in sub-clade A1 which grouped *A. arborescens* reference stains and 30 of the our *Alternaria* strains (Figure 1).

### 2.6. Design of Infectoriae Section Specific PCR Primers

Based on the deletion of 55 nucleotides observed in the first part of the obtained *gpd* sequences of *Alternaria* strains belonging to *Infectoriae* section, specific primers pair was designed by using Primer3 software [34]. In detail, forward primer (Ainf29Fw—5′CGTCTTCCGCAATGCTATCG3′) overlapped the deleted region (Figure 2) and reverse primer (Ainf277Rev—5′ACCTTGATCTCGCCCTTGAA3′) was designed in a region common to all *Alternaria* strains. PCR conditions were optimized to amplify the specific fragment of 249 nt, as follows: an initial stage at 95 °C for 2 minutes; 30 cycles each consisting of a step at 95 °C for 30 sec, 58 °C for 30 sec and 72 °C for 30 sec, and a final stage at 72 °C for 7 minutes.

### 2.7. Mycotoxins Production of Alternaria Strains

Mycotoxins, with the exception of the *A. infectoria* ITEM 17344 strain, were detected in all rice cultures, as reported in Table 3. In all the strains, AOH and AME production was very variable, ranging between 1 and 8064 mg kg^−1^ and between 1 and 14,341 mg kg^−1^ for AOH and AME, respectively. In general, as shown in Table 3, even considering the species identification obtained by phylogenetic analysis, mycotoxin production of both AOH and AME was very variable. Mean values of AOH and AME were also reported in Table 3 for each phylogenetic clade or sub-clade. Alternariol and AME production of strains belonging to sub-clade A2, showed mean values of 774 and 2099 mg kg^−1^, respectively. Similar mean values were reported for sub-clade A1 (884 and 1449 mg kg^−1^) and sub-clade A3 (651 and 1773 mg kg^−1^).

Only the strains belonging to clade B, assigned to *A. infectoria* species group, were low or no producers of toxins. The quantity of mycotoxins, with the exception of the strain ITEM17356 (AOH 223; AME 151 mg kg^−1^). ranged between 0 and 68 mg kg^−1^ (mean value 16) for AOH and between 1 and 64 mg kg^−1^ (mean value 15) for AME.

With regard to ALT production, only 2 out of 73 strains tested, *A. arborescens* ITEM 17230 (2.5 mg kg^−1^) and *A. alternata* ITEM 17244 (25 mg kg^−1^), were able to synthetize low content of this mycotoxin.

Tenuazonic acid was detected in 50 out of 73 stains with values ranging between 23 mg kg^−1^ (*A. infectoria* ITEM 17372 strain) and 3684 mg kg^−1^ (*A. mali* ITEM17349 strain). In particular, the highest TA amount was detected among the strains grouped in the sub-clade A3. All the strains belonging to A3 sub-clade showed a great capability to produce the TA with values ranging between 736 mg kg^−1^ and 3684 mg kg^−1^ (mean value 1733 mg kg^−1^). Among the strains identified as *A. alternaria* and *A tenuissima*, 8 strains were not able to produce TA and 27 out of 35 strains were high TA producers, with values ranging between 183 mg kg^−1^ and 2388 mg kg^−1^ (mean value 723 mg kg^−1^). *Alternaria arborescens* strains produced TA with values ranging between 157 and 3043 mg kg^−1^ (mean value of 730 mg kg^−1^); however 9 out of 22 strains were not a TA producer. With regard to strains grouped with *A. infectoria* reference strain, except the strain ITEM 17372 (TA content of 23 mg kg^−1^), all strains were not producers.

## 3. Discussion

In the present study, AOH contamination was detected on 82% of wheat samples with values ranging between 24 and 262 µg Kg^−1^. These data show that the occurrence of AOH in kernels collected in the field was extended and maybe it was related to the colonization process of the *Alternaria* species, as shown by Wenderoth et al. [33] in apple, citrus and tomato for *Alternaria alternata*. However, if AOH is a virulence and colonization factor for *Alternaria* infection on wheat, this needs to be confirmed. Moreover, the toxigenic risk has to be taken in account, since the AOH contamination of samples was high.

Fungicide treatments showed that *Alternaria* was less influenced by the fungicide application compared to total fungal infection. The reduction of fungicide efficacy against *Alternaria* species was previously reported by Avenot et al. [35] as a consequence of resistant strains occurring in the field. Further studies could be useful to explain the different response of fungicides against the main fungal species recovered on wheat.

*Alternaria* incidence of infected kernels, was rather lower than expected, in both experimental fields, if compared to the symptoms of black point disease. Even if black point symptoms, detected in the field, ranged from low to moderate visible infection (mean value of severity scale = 2.5), a low correlation was observed between disease symptoms and *Alternaria* infection.

This divergent result could be explained by the implication of other fungal species in the black point disease, or by abiotic damages associated with the dark brown discolouration of the wheat kernels.

The lack of congruence in *Alternaria* infection, disease symptoms and mycotoxin accumulation in wheat samples could be also linked to a different capability of the species to colonize plant tissue or to produce mycotoxins in field conditions. In addition, plant defence mechanisms, host and different pedo-climatic conditions could interfere in fungal colonization. Indeed, mycotoxin production by *Alternaria* strains revealed that almost all *Alternaria* strains tested showed a great capability to produce AOH and AME when grown in vivo on autoclaved rice, in which no mechanical barriers occur and many of the grain defences, that are thermos-labile, have been suppressed [36]. Although small-spored *Alternaria* species are of high economic relevance, they are poorly investigated among pathogenic fungi, mainly due to the difficulties of differentiating between species [37]. The morphological identification of *Alternaria* population carried out in the present study, performed according to Simmons [21], lead to assign all the strains to four main *Alternaria* species: *A. tenuissima*, *A. alternata*, *A. arborescens* and *A. infectoria*. These species are widely reported as the most occurring *Alternaria* species on wheat [8,12,14]. However, for a more accurate identification, we carried out a poly-phasic approach, using also molecular and chemical techniques [38,39].

In recent years, molecular studies revealed multiple non-monophyletic species within *Alternaria* complex and *Alternaria* species clades which do not always correlate to a species-group based upon morphological characteristics [7,14,40,41,42,43,44].

Through our phylogenetic analyses *A. alternata* and *A. tenuissima* morpho-species were not distinguishable. This result is in accordance with several studies aimed to identify *Alternaria* species by using different molecular tools, such as AFLP [45] and RAPD-PCR [46]. A wide investigation on the *Alternaria* genus by phylogenetic analysis was carried out by Woudenberg et al. [16]. Although eleven genes sequences were used, *A. alternata* and *A. tenuissima* were yet not resolved, suggesting validity of the hypothesis that considers these two morpho-species as the same phylogenetic species [16]. On the other hand, in the present study, it was possible to distinguish *A. arborescens* strains from the *A. alternata* species group, all belonging to *Alternaria* section as defined by the recent taxonomic rearrangement of *Alternaria* genus [16]. At the same section we also assigned the group, named in this work as sub-clade A3, including strains with high homology with reference strains of *A. citriarbusti*, *A. radhina* and *A. mali.*

Our results indicate that population structure of *Alternaria* species associated with black point disease, is prevalently represented by *A. tenuissima* and *A. alternata* (about 50% of total strains tested), as observed in previous studies [7,9]. Moreover, *A. arborescens* was also identified in wheat in accordance with other recent studies in Argentina [18,47], Tunisia [48], Germany, and Russia [49].

Finally, in this work, a small number of *A. infectoria* strains were also isolated from wheat, confirming a previous report from Italy [7], and other reports from Norway, Argentina, Germany and Russia [23,49,50].

Thus, *Alternaria* species composition on wheat is evolving, influenced by agricultural practices and environmental conditions. Moreover, great concern in the scientific community exists, because of the ability of *Alternaria* species to infect a huge number of field and processed wheat products and their ability to produce a range of mycotoxins, with AOH and AME being the most frequent and dangerous metabolites [12,51].

In our study, *Alternaria* strains examined showed a very variable amount of AOH and AME production, with mean values of about 800 and 1800 mg kg^−1^ for AOH and AME, respectively, except for the group (eight strains) assigned to the *A. infectoria* section. These strains showed very low toxin production abilities, confirming that *A. infectoria* strains are weak or no mycotoxin producers, as previously reported [8,45,49]. Thus, while *A. alternata*, *A. tenuissima* and *A. arborescens*, phylogenetically clustered in the same *Alternaria* section, and represent a serious risk for human heath since almost all strains produce AOH and AME [8], *A. infectoria* is not a particular threat for mycotoxin risk in wheat, due to the lack of mycotoxin production. For this reason, diagnostic tools based on the molecular detection of *A. infectoria* strains can allow a rapid evaluation of *Alternaria* mycotoxin risk assessment in wheat samples.

Consumption of wheat invaded by *Alternaria* does not necessarily imply mycotoxin presence in the grains but it indicates a potential risk for contamination. Mycotoxins of *Alternaria* have not yet received the same attention as mycotoxins produced by other fungal genera [49] although they are associated with many different diseases in humans, animals and plants [8,52]. However, nowadays consumers demand food and feed productions with a high qualitative standards to guarantee human and animal health. In this respect, and considering the continuously changing species composition, a future regulation of *Alternaria* mycotoxins could be expected, making more urgent, deeper and wider investigations on *Alternaria* species and related mycotoxins.

## 4. Materials and Methods

### 4.1. Origin of Wheat Samples

The 120 durum wheat samples analyzed in this work, were collected from two experimental fields, “San Lazzaro di Savena” (Bologna) and “Castelfranco Emilia” (Modena), located in the Emilia Romagna region, in Northern Italy. The susceptibility to black point disease of 24 durum wheat genotypes obtained with cross-breeding activities, was evaluated and compared with 6 commercial cultivars. In both fields, each genotype was grown in two replicates by using a randomized block design. In particular, one replicate was grown under high-input (fungicide application) and one replicate under low-input (no fungicide application) agronomic strategies. The treated plants were exposed to a first treatment with fungicide containing a mixture of Trifloxystrobin and Cyproconazole at the dose of 0.5 litre per hectare (L ha^−1^), at raising growth stage, and a second treatment with a formulate containing Prothioconazole, at the dose of 0.8 L ha^−1^, at earing stage. 

### 4.2. Evaluation of Black Point Symptoms

At kernels maturity stage, black point symptoms were evaluated according to an empirical scale with ten classes of severity (0 = absence of infections, 9 = extensive symptoms of infection).

### 4.3. Fungal Isolation and Growth Conditions

Representative kernels of the wheat samples (100 kernels), randomly collected at harvest time, were stored at 4 °C and then tested to detect the fungal incidence of infected kernels.

After superficial disinfection in 2% sodium hypochlorite solution for 2 min and two washing with distilled sterilized water for 1 min, 100 representative kernels for each sample, randomly collected, were plated (10 kernels/plate) on potato dextrose agar (PDA) added with 0.2 g L^−1^ of penthacloronitrobenzene (PCNB), 0.10 g L^−1^ of streptomycin sulphate salt and 0.05 g L^−1^ of neomycin. Incubation was performed at 25 ± 1 °C for 5 days under an alternating light/darkness cycle of 12 h photoperiod.

Total fungal incidence of infected kernels, and *Alternaria*, *Fusarium*, *Aspergillus* and *Penicillium* infection was calculated for each sample. Pure cultures of representative isolates belonging to *Alternaria* and *Fusarium* species, were isolated. To obtain mono-conidial isolates, conidia were spread at low density on water agar (WA: 20 g L^−1^ agar Oxoid n. 3) and singly collected after germination, using a dissection microscope, on PDA. Isolates were identified on the basis of colony and conidia morphology on PDA and potato-carrot agar (PCA: infusion from 20 g peeled and sliced white potatoes, 20 g carrot kept at 60 °C for 1 h; 15 g L^−1^ agar Oxoid n. 3), after 7 days of incubation at 25 ± 1 °C under an alternating light/darkness cycle of 12 h photoperiod, according to Simmons [35].

### 4.4. Molecular Characterization of Alternaria Strains

Ninety-two *Alternaria* strains, were selected for molecular characterization, among the set of strains isolated from wheat genotypes and commercial cultivars characterized.

The mycelium of three-days old fungal colonies, grown on cellophane disks overlaid on PDA plates, was collected by scraping and lyophilized.

Genomic DNA was extracted and purified from powdered lyophilized mycelia (10–15 mg) by using the “Wizard Magnetic DNA Purification System for Food” kit (Promega Corporation, Madison, WI), according to the manufacture’s protocol. Quantity and integrity of DNA were checked at Thermo-Scientific Nanodrop (LabX, Midland, ON, Canada) and by comparison with a standard DNA (1 kb DNA Ladder, Fermentas GmbH, St. Leon-Rot, Germany) on 0.8% agarose gel after electrophoretic separation.

Fragments of three informative genes, *alt-a1*, *gpd* and *tef*, were selected for multi-locus sequence analysis of *Alternaria* strains and amplified with the following primer pairs: alt-for/alt-rev [41], gpd1/gpd2 [53], Alt-tef1/Alt-tef2 [14], according to PCR conditions as reported in Ramires et al. [7].

Polymerase chain reaction mixture (15 μL) contained 15 ng of DNA template, 0.45 μL of each primer (10 mM), 0.3 μL of dNTPs (10mM) and 0.075 μL of Hot Master Taq DNA Polymerase (1U/μL; 5 Prime).

The PCR products were visualized with UV after electrophoretic separation in 1X TAE buffer, on 1.5% agarose gel.

Sequencing of the fragments, previously purified with the enzymatic mixture Exo/FastAP (Exonuclease I, FastAP thermosensitive alkaline phosphatase, Thermo Fisher Scientific—Waltham, MA, USA) was performed with Big Dye Terminator Cycle Sequencing Ready Reaction Kit (Applied Biosystems, Foster City, CA, USA), according to the manufacturer’s recommendations. Both strands were purified by filtration through Sephadex G-50 (5%) (Sigma-Aldrich, Aldrich, Saint Louis, MO, USA) and sequenced in “ABI PRISM 3730 Genetic Analyzer” (Applied Biosystems, Foster City, CA, USA). The FASTA sequences were obtained with BioNumerics software (Applied Maths, Kortrijk, Belgium). Phylogeny analysis was carried out including even the deposited gene sequences of the reference strains, as reported in Ramires et al. [7].

Phylogenetic trees of single and combined genes were generated by using Maximum Parsimony method and bootstrap analyses (1000 replicates, removing gaps) with MEGA5 [54].

### 4.5. Mycotoxin Extraction 

All 120 wheat samples, collected from each plot and tested for fungal infection were analyzed for the natural occurrence of AOH contamination. Only AOH occurrence was evaluated in order to select an indicator of the capability of the *Alternaria* species isolated in this study of producing mycotoxin in vivo in the field. Chemical analyses on the 92 *Alternaria* strains, molecularly characterized, were also carried out to evaluate the capability of *Alternaria* strains to produce AOH, AME, ALT and TA. According to Li et al. [55], all the strains were inoculated on sterilized polished rice and grown for 21 days at 25 ± 1 °C in darkness. The cultures were then dried and powdered before mycotoxin extraction.

The method used for mycotoxins analysis is based on that described by Rubert et al. [56] with some modifications. The samples were finely ground with an Oster Classic grinder (220–240V, 50/60 Hz, 600W; Madrid, Spain). Five grams of each homogenized sample was weighed in a 50 mL plastic tube and 25 mL of methanol was added. The extraction was carried out using an Ultra Ika T18 basic Ultra-turrax (Staufen, Germany) for 3 min. The extract was centrifuged at 4000 rpm for 5 min at 5 °C. One mL of the surnatant was filtered through a 13mm/0.22 μm nylon filter and diluted before injection into high performance liquid chromatography associated with a diode array detector (LC-DAD). All the extractions were carried out in triplicate.

#### HPLC Analysis

Alternariol, AME, ALT and TA were determined using Merk HPLC with a diode array detector (LC-DAD) L-7455 (Merk, Darmstadt, Germany) at 256 nm and Hitachi Software Model D-7000 version 4.0 was used for data analysis. A Gemini C18 column (Phenomenex, Torrance, USA) 4.6 × 150 mm, 3 μm particle size was used as the stationary phase. The mobile phase consisted of two eluents, namely, eluent A (water with 50 µL/L trifluoroacetic acid) and eluent B (acetonitrile with 50 µL/L trifluoroacetic acid). A gradient program with a constant flow rate of 1 ml/min was used, starting with 90% A and 10% B, reaching 50% B after 15 min and 100% B after 20 min. 100% B was maintained for 1 min. Thereafter the gradient was returned to 10% B in 1 min and allowed to equilibrate for 3 min before the next analysis [57].

## Figures and Tables

**Figure 1 toxins-12-00275-f001:**
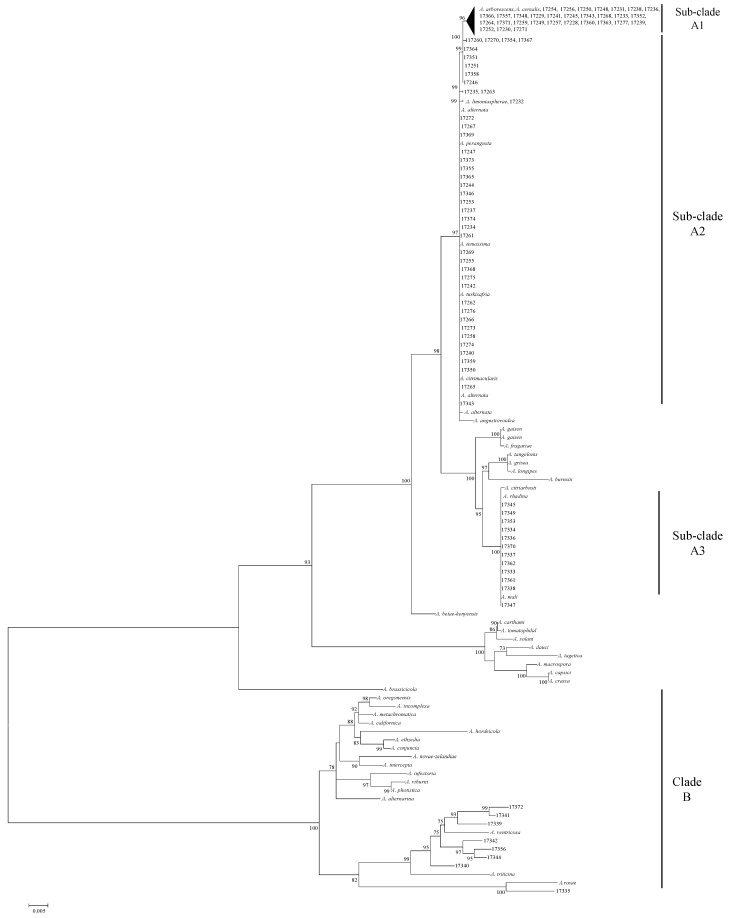
Phylogenetic tree generated by maximum parsimony method (bootstrap 1000 replicates) of combined allergen alt a1 (*alt-a1*), glyceraldeyde-3-phosphate dehydrogenase (*gpd*), and translation elongation factor 1-α (*tef*) gene sequences of 92 *Alternaria* strains.

**Figure 2 toxins-12-00275-f002:**
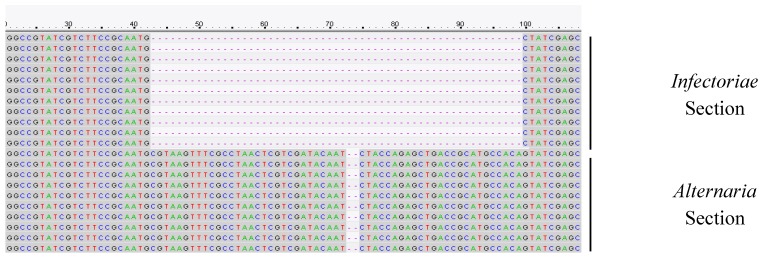
Deletion of 55 nt in *gpd* sequence of *Alternaria* strains belonging to *Infectoriae* section. Ainf29Fw primer (5′-CGTCTTCCGCAATGCTATCG3-3′), designed on deletion site, was able to discriminate among *Alternaria* and *Infectoriae* Sections.

**Table 1 toxins-12-00275-t001:** Blackpoint symptoms assessed in Bologna experimental field, *Alternaria* incidence of infected kernels and alternariol contamination on durum wheat genotypes.

Sample Code	Genotype	Treated with Fungicides *			Untreated		
		Blackpoint Symptoms (0–9 Scale)	Alternaria Incidence of Infected Kernels (%)	Alternariol (µg Kg^−1^)	Blackpoint Symptoms (0–9 Scale)	Alternaria Incidence of Infected Kernels (%)	Alternariol (µg Kg^−1^)
G1	D04802	4.0	5	34	5.0	0	87
G2	D05422	2.5	7	72	4.5	12	148
G3	508Gd05/101	1.0	6	24	4.0	12	101
G4	508gd05/209	0.5	8	31	1.5	11	Nd
G5	508gd07/10T	0.5	10	Nd	0.5	8	109
G6	D07422	1.0	2	47	3.0	4	88
G7	909gd08/20B	0.5	6	44	2.0	11	109
G8	909gd08/77	2.5	5	79	6.0	13	Nd
G9	909gd08/94A	6.0	5	Nd	8.0	16	124
G10	905gd09/37	2.0	15	55	4.0	9	115
G11	505gd09/131A	1.5	12	51	3.5	8	-
G12	505gd09/131B	3.0	26	31	4.5	5	84
G13	906gd10/4T	3.0	16	37	4.0	7	85
G14	906gd104P	3.5	8	65	4.0	14	112
G15	906gd10/21A	2.0	4	44	4.0	13	77
G16	906gd10/21B	4.0	10	36	6.0	8	97
G17	906gd10/40T	0.0	5	27	1.5	8	88
G18	906gd10/40P	0.5	11	85	1.0	6	98
G19	906gd10/72T	1.5	0	39	3.5	9	106
G20	906gd10/72P	1.5	8	40	3.0	3	89
G21	906gd010/88A	1.0	0	43	1.5	5	77
G22	906gd10/88B	0.5	5	26	3.0	8	40
G23	906gd10/118T	0.5	3	Nd	1.5	12	93
G24	906gd10/118P	1.5	1	28	4.0	16	93
**Min value**		**0**	**0**	**0**	**0.5**	**0**	**0**
**Max value**		**6**	**26**	**85**	**8**	**16**	**148**
**Average**		**1.9**	**7.4**	**39.1**	**3.5**	**9.1**	**84.2**
**Standard Deviation**		**1.5**	**5.8**	**22.3**	**1.8**	**4.1**	**34.3**
G25	CLAUDIO	1.0	6	42	6.0	9	97
G26	IRIDE	3.0	9	91	6.0	6	Nd
G27	LIBERDUR	1.0	20	44	2.0	11	77
G28	MARCO AURELIO	3.0	2	66	5.0	21	110
G29	MIRADOUX	1.0	7	87	2.0	10	87
G30	SARAGOLLA	2.5	9	92	4.0	17	110

* First treatment with trifloxystrobin+cyproconazole (0.5 Lha^−1^) during doffing time and a second treatment with prothioconazole (0.8 Lha^−1^) during earing time.

**Table 2 toxins-12-00275-t002:** Black point symptoms assessed in Modena experimental field, *Alternaria* incidence of infected kernels and alternariol contamination on durum wheat genotypes.

Sample Code	Genotype	Treated with Fungicides *			Untreated		
		Blackpoint Symptoms (0–9 Scale)	*Alternaria* Incidence of Infected Kernels (%)	Alternariol (µg Kg^−1^)	Blackpoint Symptoms (0–9 Scale)	*Alternaria* Incidence of Infected Kernels (%)	Alternariol (µg Kg^−1^)
G1	D04802	3.0	8	118	5.0	14	118
G2	D05422	6.0	11	141	6.0	15	Nd
G3	508Gd05/101	1.5	13	124	3.0	13	133
G4	508gd05/209	0.5	6	97	1.0	11	Nd
G5	508gd07/10T	0.5	4	262	1.0	12	105
G6	D07422	0.5	7	161	1.5	9	230
G7	909gd08/20B	1.0	6	76	2.0	12	Nd
G8	909gd08/77	5.0	10	147	6.0	14	84
G9	909gd08/94A	2.0	6	64	3.5	13	Nd
G10	905gd09/37	0.5	2	Nd	1.0	10	97
G11	505gd09/131A	1.0	6	100	3.0	12	Nd
G12	505gd09/131B	1.0	4	100	2.0	12	Nd
G13	906gd10/4T	2.0	11	111	3.5	5	120
G14	906gd104P	1.0	12	127	4.0	13	Nd
G15	906gd10/21A	1.5	17	111	4.0	15	Nd
G16	906gd10/21B	5.0	6	107	3.0	14	95
G17	906gd10/40T	1.0	8	180	2.0	12	Nd
G18	906gd10/40P	0.5	5	Nd	2.0	12	Nd
G19	906gd10/72T	1.0	9	104	5.0	14	125
G20	906gd10/72P	0.5	7	Nd	1.5	12	68
G21	906gd010/88A	2.0	4	91	4.0	15	94
G22	906gd10/88B	2.0	10	111	3.0	3	171
G23	906gd10/118T	0.5	3	139	1.0	5	78
G24	906gd10/118P	0.5	7	141	2.0	11	71
**Min value**		**0.5**	**2**	**0**	**1**	**3**	**0**
**Max value**		**6**	**17**	**262**	**6**	**15**	**230**
**Average**		**1.7**	**7.6**	**108.8**	**2.9**	**11.6**	**66.2**
**Standard Deviation**		**1.6**	**3.5**	**57.3**	**1.5**	**3.2**	**65.8**
G25	CLAUDIO	0.5	9	76	3.0	3	Nd
G26	IRIDE	0.5	5	71	2.0	8	155
G27	LIBERDUR	1.0	9	157	1.5	17	83
G28	MARCO AURELIO	2.0	7	109	4.0	5	97
G29	MIRADOUX	0.5	16	Nd	1.0	20	160
G30	SARAGOLLA	1.0	10	Nd	2.0	13	169

* First treatment with trifloxystrobin+cyproconazole (0.5 Lha^−1^) during doffing time and a second treatment with prothioconazole (0.8 Lha^−1^) during earing time.

**Table 3 toxins-12-00275-t003:** Alternariol (AOH), alternariol methyl ether (AME), altenuene (ALT) and tenuazonic acid (TE) content, produced by *Alternaria* strains.

Strain (ITEM)	mg kg^−1^				Strain (ITEM)	mg kg^−1^				Strain (ITEM)	mg kg^−1^			
	AOH	AME	ALT	TA		AOH	AME	ALT	TA		AOH	AME	ALT	TA
**17235 ^A2^**	67	182	-	-	**17272 ^A2^**	38	310	Nd	1030	**17348 ^A1^**	127	501	Nd	1775
**17237 ^A2^**	815	2618	Nd	603	**17274 ^A2^**	46	185	Nd	965	**17271 ^A1^**	448	2703	-	-
**17242 ^A2^**	1070	2557	Nd	1022	**17276 ^A2^**	26	353	Nd	977	**17277 ^A1^**	1085	2113	Nd	311
**17243 ^A2^**	251	2167	Nd	306	**17350 ^A2^**	65	371	Nd	476	**17352 ^A1^**	1002	1306	Nd	157
**17244 ^A2^**	100	1000	25	Nd	**17351 ^A2^**	69	798	Nd	934	**17357 ^A1^**	752	3849	Nd	2725
**17247 ^A2^**	1682	2349	-	-	**17354 ^A2^**	85	9	Nd	1596	**17360 ^A1^**	1	5	-	-
**17258 ^A2^**	46	65	-	575	**17355 ^A2^**	372	2239	-	-	**17363 ^A1^**	145	14	Nd	Nd
**17260 ^A2^**	1169	7965	Nd	755	**17358 ^A2^**	138	369	Nd	572	**17366 ^A1^**	22	30	Nd	Nd
**17269 ^A2^**	378	129	Nd	Nd	**17364 ^A2^**	872	2396	-	-	**17371 ^A1^**	1512	2035	Nd	Nd
**17270 ^A2^**	872	50	Nd	Nd	**17374 ^A2^**	992	3357	Nd	642	**Average ^A1^**	**884**	**1449**	**-**	**730**
**17273 ^A2^**	1822	5603	Nd	1654	**Average ^A2^**	**774**	**2099**	**-**	**723**	**17333 ^A3^**	1633	1934	-	-
**17275 ^A2^**	7372	13,156	Nd	1009	**17228 ^A1^**	651	548	Nd	Nd	**17334 ^A3^**	60	91	-	-
**17346 ^A2^**	3152	7143	Nd	938	**17229 ^A1^**	23	33	-	-	**17336 ^A3^**	62	620	Nd	2143
**17359 ^A2^**	11	5	-	-	**17230 ^A1^**	4176	4213	2.5	822	**17337 ^A3^**	415	2866	-	-
**17365 ^A2^**	3581	14,341	Nd	1656	**17231 ^A1^**	1283	2567	Nd	Nd	**17338 ^A3^**	1781	2763	Nd	1084
**17367 ^A2^**	62	39	Nd	Nd	**17233 ^A1^**	22	163	Nd	3043	**17345 ^A3^**	28	385	Nd	2485
**17368 ^A2^**	40	374	Nd	377	**17236 ^A1^**	1283	1612	-	-	**17347 ^A3^**	213	288	Nd	1767
**17369 ^A2^**	57	230	-	-	**17238 ^A1^**	925	1833	-	-	**17349 ^A3^**	73	1039	Nd	3684
**17373 ^A2^**	245	823	Nd	985	**17239 ^A1^**	89	10	Nd	Nd	**17353 ^A3^**	451	287	Nd	773
**17232 ^A2^**	1962	4485	Nd	Nd	**17241 ^A1^**	154	29	Nd	Nd	**17361 ^A3^**	23	16	Nd	736
**17234 ^A2^**	1296	2775	Nd	Nd	**17245 ^A1^**	8064	8597	-	-	**17362 ^A3^**	636	2261	Nd	2111
**17246 ^A2^**	842	1257	Nd	914	**17248 ^A1^**	1309	1067	Nd	1417	**17370 ^A3^**	2434	8730	Nd	816
**17251 ^A2^**	34	30	Nd	710	**17249 ^A1^**	948	2606	Nd	920	**Average ^A3^**	**651**	**1773**		**1733**
**17253 ^A2^**	32	127	Nd	2059	**17250 ^A1^**	531	2437	Nd	750	**17335 ^B^**	68	4	Nd	Nd
**17255 ^A2^**	162	605	Nd	621	**17252 ^A1^**	611	787	-	-	**17339 ^B^**	25	64	Nd	Nd
**17261 ^A2^**	221	2174	-	-	**17254 ^A1^**	50	53	Nd	Nd	**17340 ^B^**	12	19	Nd	Nd
**17262 ^A2^**	25	6	Nd	901	**17256 ^A1^**	505	1045	Nd	953	**17341 ^B^**	8	11	-	-
**17263 ^A2^**	Nd	49	Nd	183	**17257 ^A1^**	Nd	Nd	Nd	605	**17342 ^B^**	5	4	Nd	Nd
**17240 ^A2^**	316	166	Nd	Nd	**17259 ^A1^**	240	87	Nd	680	**17344 ^B^**	Nd	Nd	Nd	Nd
**17265 ^A2^**	44	357	Nd	2388	**17264 ^A1^**	112	152	Nd	1911	**17356 ^B^**	223	151	Nd	Nd
**17266 ^A2^**	420	2178	Nd	463	**17268 ^A1^**	300	635	Nd	Nd	**17372 ^B^**	0	1	Nd	23
**17267 ^A2^**	880	2765	Nd	Nd	**17343 ^A1^**	143	2442	-	-	**Average ^B^**	**43**	**32**	**-**	**-**

A2 = *A. alternata/A. tenuissma* (sub-clade A2); A1 = *A. arborescens* (sub-clade A1); A3 = *A. mali* (sub-clade A3); B = *A. infectoria* (clade B); Nd = Not detected.

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
