# Peer review of "Molecular Identification and Mycotoxin Production by Alternaria Species Occurring on Durum Wheat, Showing Black Point Symptoms"

_toxins, 2020, doi:10.3390/toxins12040275_

Round 1
Reviewer 1 Report
The authors, who cannot be found in the paper, have done an extensive study and data collection for certain chemical compositions associated with mycotoxin production of Alternaria species in 120 durum wheat samples. Unfortunately, the construction of the paper makes it a difficult task for readers to extract meaningful information from exhaustive data listed in the tables. Both tables (1 & 2) show no correlation of black point symptoms with either fungi contamination or mycotoxin concentration, which raises the question of what rating standards of the black point symptoms (0-9 scale) were based on. The correlation between contamination and mycotoxin is weak as well. There is an interesting finding about low toxin production in A. infectoria species infected groups but which has been previously reported. There is no discussion for the figure 2 showing the deletion of 55 nucleotides in gpd sequence in A. infectoria species, which could be responsible for the low toxins production and deserves a thorough investigation. The two and a half pages of the discussion section is redundant and poorly organized. The first 5 paraphs merely repeated the background information and the rest part was focused on the discussion of Alternaria classification between morphological and phylogenetic analysis, which is not quite related to the study presented. Overall, the work has important value of studying the toxin production by Alternatia for an arrange of genotypes of durum wheat but a serious revision is recommended.
Author Response
The authors, who cannot be found in the paper, have done an extensive study and data collection for certain chemical compositions associated with mycotoxin production of Alternaria species in 120 durum wheat samples. Unfortunately, the construction of the paper makes it a difficult task for readers to extract meaningful information from exhaustive data listed in the tables. Both tables (1 & 2) show no correlation of black point symptoms with either fungi contamination or mycotoxin concentration, which raises the question of what rating standards of the black point symptoms (0-9 scale) were based on.
The standards of the black point symptoms scale have been described in material and methods: 0 for absence of symptoms – 9 for full symptoms occurrence on the ear. The 10 classes were arbitrarily established as always breeders do for their disease evaluation assessment in field
The correlation between contamination and mycotoxin is weak as well. There is an interesting finding about low toxin production in A. infectoria species infected groups but which has been previously reported.
We think that this confirmation is important also since the strains have been isolated from other geographical areas compared the previous reports
There is no discussion for the figure 2 showing the deletion of 55 nucleotides in gpd sequence in A. infectoria species, which could be responsible for the low toxins production and deserves a thorough investigation.
Glyceraldehyde-3-phosphate dehydrogenase is an informative housekeeping gene, not related to mycotoxin production. It is largely used to study the phylogenetic relationship and the genetic variability within and among Alternaria species. In this work, the sequence analyses of Alternaria strains has allowed us to observe a deletion of 55 nt in all strains belonging to Infectoria Section. Since Alternaria species belonging to Infectoria section are not able to produce the most important Alternaria mycotoxins, we set up a molecular protocol for the rapid detection of non mycotoxin producing Alternaria species.
The two and a half pages of the discussion section is redundant and poorly organized. The first 5 paraphs merely repeated the background information and the rest part was focused on the discussion of Alternaria classification between morphological and phylogenetic analysis, which is not quite related to the study presented.
We totally agree with these remarks, therefore, we have almost completely re-written the introduction following the suggestion of the referee, we reduced a lot the discussion deleting redundant parts, and we reduced a lot the part dealing with the phylogenetic vs molecular analyses, which at least a bit needs to be reported due to the continuous change in taxonomy on this genus.
Overall, the work has important value of studying the toxin production by Alternatia for an arrange of genotypes of durum wheat but a serious revision is recommended.

Reviewer 2 Report
The manuscript is interesting from the point of view of the different studies carried out to establish a relationship between black points, fungi and mycotoxins produced.
However, in general, the structure of the article does not allow us to understand the conclusions drawn from the results and there are decisions that are not detailed, so the weight of the decisions taken is lost. The most relevant aspects that should be corrected by the authors for the acceptance of the article will be detailed below. Therefore, it is necessary to make major changes before it can be accepted.
Specific comments
The abstract does not show essential information that should go into this part. Although the interest of the study and the most relevant results are detailed, it is not clear what study is done at the analytical level and the techniques used. The problem with the abstract is that it does not allow us to know the content of the article, which diminishes its attractiveness. In addition, the abbreviations of the different mycotoxins are detailed but then not applied, so the authors should unify this nomenclature.
The introduction does not provide a detailed background to this study, but it does appear in the conclusions. Therefore, the authors should restructure the information so that the importance and novelty of the work can be understood.
In the results detailed in Tables 1 and 2, the most relevant results are shown but the uncertainty of the method is not specified (which is also not detailed in table 3), neither because only the alternariol content is evaluated nor on the basis that the scale is applied to the black point symptoms.
In addition, these studies compare samples treated with a fungicide with those not treated. The problem is that the authors do not detail the interest of this comparison nor do they specify if it is important in the objective of the work.
As discussed above, the first paragraphs of the conclusion detail background information that would be most useful if incorporated into the introduction. In addition, this section details the results but does not specify or draw any conclusions from them.
Finally, the HPLC method does not specify the conditions under which the different analytes are observed since some of them, when analyzed by DAD, require the addition of metal ions in order to be detected. Although it is not the important thing in the article it could be useful to see a chromatogram of the four analytes determined in the same sample.
Author Response
Dear Editor,
we thank the great revisions of our paper by the referees which work was very valuable. We tried to reply to their remarks as reported below.
The manuscript is interesting from the point of view of the different studies carried out to establish a relationship between black points, fungi and mycotoxins produced.
However, in general, the structure of the article does not allow us to understand the conclusions drawn from the results and there are decisions that are not detailed, so the weight of the decisions taken is lost. The most relevant aspects that should be corrected by the authors for the acceptance of the article will be detailed below. Therefore, it is necessary to make major changes before it can be accepted.
Specific comments
The abstract does not show essential information that should go into this part. Although the interest of the study and the most relevant results are detailed, it is not clear what study is done at the analytical level and the techniques used. The problem with the abstract is that it does not allow us to know the content of the article, which diminishes its attractiveness. In addition, the abbreviations of the different mycotoxins are detailed but then not applied, so the authors should unify this nomenclature.
We change deeply the abstract. We added the equipment used for the chemical analyses, that is reported in Material and Methods. We did not understand the remark on the abbreviations of mycotoxins. We used them along the whole abstract and also along the paper. We don’t use them only after the dot, as required by the Journal style.
The introduction does not provide a detailed background to this study, but it does appear in the conclusions. Therefore, the authors should restructure the information so that the importance and novelty of the work can be understood.
We totally agree with these remarks, therefore, we have almost completely re-written the introduction following the suggestion of the referee
In the results detailed in Tables 1 and 2, the most relevant results are shown but the uncertainty of the method is not specified (which is also not detailed in table 3), neither because only the alternariol content is evaluated nor on the basis that the scale is applied to the black point symptoms.
The standards of the black point symptoms scale have been described in material and methods: 0 for absence of symptoms – 9 for full symptoms occurrence on the ear. The 10 classes were arbitrarily established as always breeders do for their disease evaluation assessment in field; we also added a sentence explaining why we used Alternariol as mycotoxin to be analyzed
In addition, these studies compare samples treated with a fungicide with those not treated. The problem is that the authors do not detail the interest of this comparison nor do they specify if it is important in the objective of the work.
We discussed this between lines 270-275” “Moreover, fungicide treatments were effective to contain total fungal contamination, although the Alternaria contamination seems to be slightly influenced by fungicide efficacy. The reduction of fungicide activity against Alternaria species was previously reported by Avenot et al., [48] as consequence of occurring resistant field strains. Further studies could be useful to explain the different response of fungicides against the main fungal species recovered on wheat.”
As discussed above, the first paragraphs of the conclusion detail background information that would be most useful if incorporated into the introduction. In addition, this section details the results but does not specify or draw any conclusions from them.
We totally agree with these remarks, therefore, we have almost completely re-written the introduction following the suggestion of the referee, we reduced a lot the discussion deleting redundant parts, and we reduced a lot the part dealing with the phylogenetic vs molecular analyses, which at least a bit needs to be reported due to the continuous change in taxonomy on this genus..
Finally, the HPLC method does not specify the conditions under which the different analytes are observed since some of them, when analyzed by DAD, require the addition of metal ions in order to be detected. Although it is not the important thing in the article it could be useful to see a chromatogram of the four analytes determined in the same sample.
The method used for the identification and quantification of the mycotoxins studied are detailed in the paragraph 4.5.1. We used the same method published in the article indicated below where are indicated all the conditions used for the method application and validation.
"Myresiotis, C.K.; Testempasis, S.; Vryzas, Z.; Karaoglanidis, G.S.; Papadopoulou-Mourkidou, E. Determination of mycotoxins in pomegranate fruits and juices using a QuEChERS-based method. Food Chem. 2015, 182, 81-88. doi: 10.1016/j.foodchem.2015.02.141".
We didn't add in the text a chromatogram because the analysis of the metabolites are expressed in the tables. The chromatogram is just a graph, and actuality is added just in the pure analytical papers

Round 2
Reviewer 2 Report
The authors have followed the recommendations of the reviewers and succeeded in solving the problems detailed in the previous reports. For this reason I believe that the article should be accepted as it stands in this second review.
Author Response
Thank you